# SiC$_p$/Al5056 Composite Coatings Applied to A Magnesium Substrate by Cold Gas Dynamic Spray Method for Corrosion Protection

**Yingying Wang** [1]**, Bernard Normand** [2,]***, Hanlin Liao** [3]**, Guofeng Zhao** [1]**, Nicolas Mary** [2] **and Junlei Tang** [1,]***

[1] College of Chemistry and Chemical Engineering, Southwest Petroleum University, Chengdu 610500, China; yingyingwanglyon@126.com (Y.W.); zhaoguofeng67@163.com (G.Z.)

[2] Université de Lyon, INSA-Lyon, MATEIS CNRS UMR 5510, 69 621 Villeurbanne cedex, France; nicolas.mary@insa-lyon.fr

[3] ICB UMR 6303, CNRS, University Bourgogne Franche-Comté, UTBM, 90100 Belfort Cedex, France; hanlin.liao@utbm.fr

* Correspondence: bernard.normand@insa-lyon.fr (B.N.); tangjunlei@126.com (J.T.)

**Abstract:** Corrosion protection using cold spraying is a promising method to address the shortcomings associated with classical techniques for protecting magnesium alloys from corrosion. In this study, SiC$_p$/Al 5056 composite coatings were prepared on a magnesium substrate using cold spraying. The effects on the microstructure and corrosion properties after adding SiC were analysed. To evaluate the durability of the cold-sprayed Al-based coatings on Mg, galvanic corrosion, immersion and thermal cycling tests were conducted. The results show that cold-sprayed aluminium coatings serve as a reliable cathode for magnesium substrates. The addition of SiC particles increases the galvanic potential and decreases the galvanic reduction current of the coating/substrate couple. The SiC$_p$/Al 5056 composite coatings show better corrosion resistance than that of the Al 5056 coating in extended immersion tests due to the densification of the coating under the peening effect of hard particles. Moreover, SiC particles with an average size of 15.6 μm show more improvement than with SiC particles having an average size of 72.8 μm. The cold-sprayed SiC$_p$/Al 5056 composite coating also presents excellent properties in the thermal cycling tests. After applying failure mode parameters in the thermal cycling tests, the composite coating demonstrates good adhesion as cracking was located in the Mg substrate and not at the interface.

**Keywords:** cold gas dynamic spray; SiC$_p$/Al 5056 composite coatings; corrosion protection; galvanic corrosion; thermal cycling

## 1. Introduction

Magnesium and its alloys are widely used in aerospace, automotive and other industries because of their good strength-to-weight and strength-to-cost ratios [1]. Nevertheless, the engineering applications of Mg alloys are mainly limited by their poor corrosion resistance. The undesirable corrosion properties of Mg alloys come from their high electrochemical activity [2], and many alloying elements negatively impact the overall corrosion resistance, e.g., copper and nickel [3]. Furthermore, pure Mg has the lowest corrosion potential (approximately −2.37 V vs. a standard hydrogen electrode (ENH)) among all structural metals, which makes Mg alloys suffer from serious galvanic corrosion when coupled with another metal. Several studies have led to the emerging of new protection techniques for Mg alloys [4]. Aluminium coatings are a good alternative to Mg corrosion protection due to their low potential gap, density difference and economic cost. The application of an Al coating to a Mg alloy

requires a process that provides good corrosion protective capabilities of the coatings and no adverse effects on the Mg base. Among the various surface techniques used to prolong the service life of Mg alloys, such as thermal spraying [5,6], anodizing [7,8], plasma electrolytic oxidation treatment [9,10], physical vapor deposition [11,12], plasma immersion ion implantation [13,14], and laser surface modification [15,16], cold spraying is a promising, simple and environmentally friendly solution [17]. Both pure Al coatings [18] and Al-based composite coatings [19] have been successfully produced on Mg and Mg alloys using cold spraying.

Cold spraying, also called cold gas dynamic spray [20], high-velocity particle consolidation [21], supersonic particle deposition [22] or kinetic spraying [23], is an emerging coating technology in the family of thermal spraying. Compared with conventional thermal spray processes, a distinguishing feature of cold spraying is the ability to produce coatings with gas temperatures much lower than the melting point of the deposited materials. As a consequence, the deleterious effects of oxidation, evaporation and debonding, residual stresses, gas releasing problems and undesirable phase changes under high-temperature, along with any other concerns associated with liquefaction-solidification processes during thermal spraying, are minimized or eliminated [24]. For these reasons, cold spraying is a very attractive method to deposit oxidation-sensitive alloys such as aluminium alloys, magnesium alloys and titanium alloys to elaborate thick and dense coatings.

The usefulness of cold-sprayed coatings for controlling the corrosion resistance of industrial components has been investigated. In contrast to studying corrosion performance, investigations on cyclic oxidation conditions [25] and cyclic loads [18] have not received much attention. These performance investigations are of fundamental importance when exploring the possible applications of cold-sprayed coatings or when optimizing cold spraying techniques [26]. Tao et al. [19] reported that the addition of $\alpha$-Al$_2$O$_3$ in Al matrix leads to high adhesion to a substrate and a high tensile strength but does not have an obvious influence on the corrosion current densities. Diab et al. [18] found that a pure Al coating on an AZ31B alloy demonstrated strong adhesion and excellent corrosion performance. However, exposure to a corrosive environment significantly decreased the fatigue life of the coating because of pitting and cracking [18].

Dense Al coatings and Al-based composite coatings with good corrosion resistance were obtained on Al-alloy substrates in our previous work [27–30]. The authors found that adding SiC ceramic particles provided beneficial effects, such as a decrease of coating porosity; moreover, the average hardness of the composite coatings increased with increasing additions of SiC particles. Similar to the results in [25], the addition of SiC particles had no influence on the corrosion current densities in electrochemical measurements. However, long-term immersion tests demonstrated that the Al-composite coatings had better corrosion resistances than that of the Al 5056 alloy coating [27,28].

The present work focused on the feasibility of applying corrosion resistant Al-based coatings on Mg using cold spraying. Two Al 5056-based composite coatings were selected from previous investigations. The composite coatings contained 30 vol.% SiC particles as reinforcement [28,29], with average sizes of the SiC particles being 15.6 $\mu$m and 72.8 $\mu$m [28,30]. An unreinforced Al 5056 coating is deposited as a reference. The corrosion protection mechanism of the Al-based coatings on Mg was completely different than that observed for an Al substrate covered with Al coatings. In this latter case, the coating provided cathodic protection. Since Al-based coatings exhibited a higher potential than their Mg substrates, the galvanic effect was deleterious for the Mg substrate. No connective porosity could be accepted in this case. Considering the coefficient of the thermal expansion mismatch between the Al coating and the Mg substrate, thermal cycling tests were also conducted to evaluate the durability of the Al-based cold-sprayed coatings on Mg under thermal stresses.

## 2. Materials and Methods

### 2.1. Materials

Commercially available magnesium (Mg ≥ 99 wt.%) plates (50 mm × 50 mm) were used as the substrate material. Gas-atomized Al 5056 powder (ICB, Belfort, France) and commercially available crushed SiC powder (H.C. Starck, Newton, MA, USA) composed the feedstock. The feedstock mixtures of aluminium alloy 5056 powder and 30 vol.% SiC powder were deposited using cold spraying. The particle size was analysed with laser diffractometry (Mastersizer 2000, Malvern Instruments Ltd., Malvern, UK). The nominal chemical composition (wt.%) of the spherical Al 5056 powder was Al–5.0 Mg–0.12 Mn–0.12 Cr. The average SiC particle sizes were 15.6 μm and 72.8 μm, giving the composite coatings referenced as Comps1 and Comps2 respectively. More details about the morphology and size distribution of the Al 5056 particles and SiC particles can be found in [30].

### 2.2. Cold Spraying Procedure

Cold spraying was performed with a commercial spray gun (Kinetic 3000, CGT GmbH, Asbach, Germany). Parameters used for this study are the same as those used for previous work described in [28,30]. Compressed air was used as a driving gas at a pressure of 2.6 MPa and a temperature of 600 °C. The standoff distance was fixed at 30 mm.

### 2.3. Coating Characterization

The metallographic samples were prepared as follows: first, they were embedded in hot resin (Mecapress I, Presi, Grenoble, France), ground using SiC abrasive papers from P180 to P1200 and polished using 1 μm of graded alumina suspensions. Field emission scanning electron microscopy (FE-SEM, Supra® 55 VP, Carl Zeiss NTS GmbH, Oberkochen, Germany) was used to measure the coating thickness and porosity and to observe the microstructure on a cross-section of the coating. The procedure to evaluate porosity is described in [28,30].

The microhardnesses of the coatings were tested using a Vickers microhardness indenter (Leitz, Germany) with a load of 300 g and a dwell time of 15 s on the polished cross-sections. Ten indentations at random locations were averaged to determine the coating hardness.

Corrosion behaviours were evaluated by performing electrochemical measurements and extended immersion tests in an aerated 0.1 M sodium sulfate ($Na_2SO_4$) solution. Electrochemical measurements were performed using a standard three-electrode electrochemical setup on an electrochemical workstation (Reference 600[TM], Potentiostat/Galvanostat/ZRA, Gamry Instruments, Inc., Warminster, PA, USA). The electrochemical measurements were repeated at least twice to confirm their reproducibility.

Open circuit potential (OCP) measurements were conducted over 12 h. Potentiodynamic polarization scans were performed from −0.25 V to 1.5 V (vs. OCP) with a scan rate of 1 mV/s. Electrochemical impedance spectroscopy (EIS) was performed at OCP in a frequency range from $10^5$ Hz to $10^{-2}$ Hz and with a sinusoidal signal perturbation of 10 mV. During the extended immersion tests, the OCP and EIS were recorded at different times.

Each electrochemical measurement allowed several electrochemical criteria to be highlighted, ensuring the classification of the coating performances [31].

For the galvanic corrosion measurements, a specific experiment was performed. The coating was connected in parallel to the substrate. The cold-sprayed coating was connected as the working electrode (WE), and a Mg plate was connected as the counter electrode (CE). The galvanic current and galvanic potential were monitored over 24 h using a zero-resistance ammeter (ZRA) technique. To detect the effect of stirring, a magnetic stirrer was used discontinuously during the 48-h galvanic corrosion measurements.

To evaluate the durability of the cold-sprayed $SiC_P$/Al 5056 composite coating on a Mg substrate, thermal cycling tests were conducted. Two thermal cycling conditions were used: I) 10 min of heating at 400 °C, while buried in $SiO_2$ grits in a furnace pot, and 3 min of cooling at 0 °C, while immersed

in a beaker of ice water and II) 10 min of heating at 200 °C, while immersed in dimethyl silicone (Analytical Reagent, rational formula $(CH_3)_3SiO[Si(CH_3)_2O]_nSi(CH_3)_3$), and 3 min of cooling at 0 °C, while immersed in a beaker of ice water. The microstructures of the damaged coatings from the thermal shock were investigated using scanning electron microscopy.

## 3. Results

### 3.1. Microstructure

Figure 1 shows the top surface morphology of the Al 5056, Comps1 and Comps2 coatings. Similar to the case of aluminium substrate, the cold-sprayed aluminium coatings exhibit a rough outer surface on the magnesium substrate [28]. The majority of the deposited particles on the top layer adhere well. The porous top layer is due to the absence of the shot peening effect of the next incoming particles. Crevices can be detected beside the loosely adhered particles. Nonetheless, the addition of ceramic particles reduces the number of crevices as the particles undergo more intensive plastic deformation.

Figure 2 shows the full-thickness, cross-sectional SEM micrographs of the cold-sprayed Al 5056 coating and SiC_p/Al 5056 composite coatings. The dark region on the top is the resin used to embed the sample, the uniformly grey area at the bottom is the Mg substrate, and the area between these two regions is the cold-sprayed coating. The pores can be identified as dark dots.

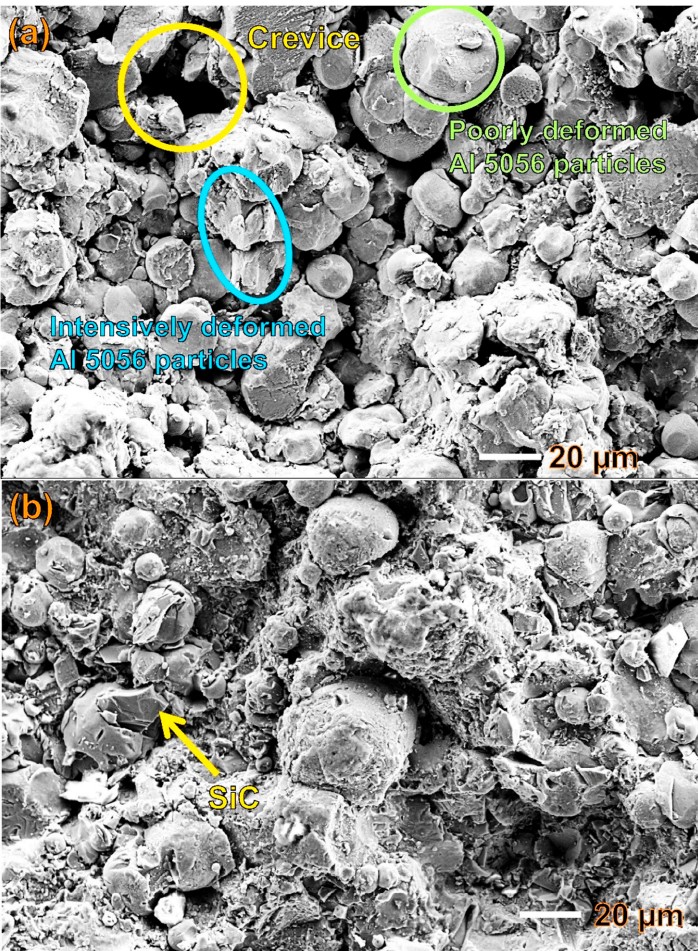

**Figure 1.** *Cont.*

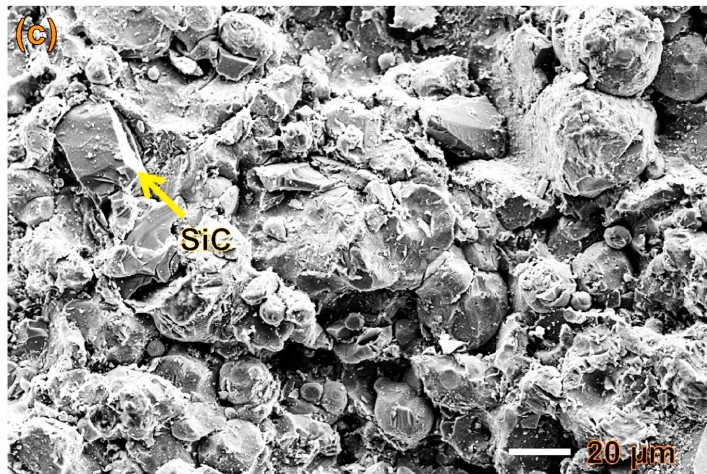

**Figure 1.** Top surfaces of the (**a**) cold-sprayed Al 5056, (**b**) Comps1 and (**c**) Comps2 coatings.

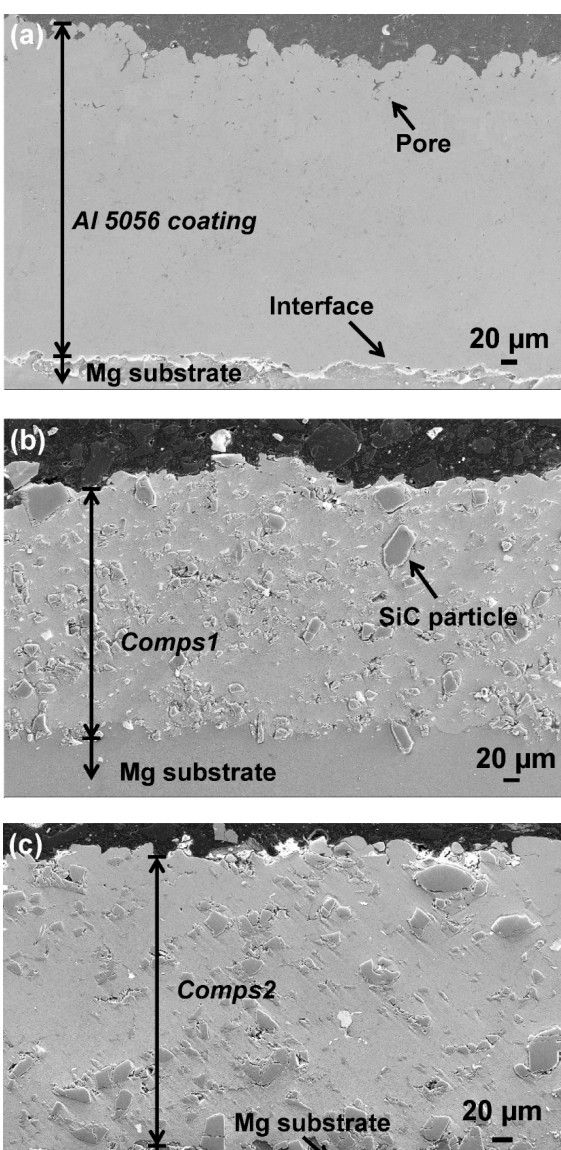

**Figure 2.** Cross-sectional SEM micrographs of the Al 5056 coating (**a**) and SiC$_p$/Al 5056 composite coatings reinforced by different sizes of SiC particles: (**b**) Comps1 coating and (**c**) Comps2 coatings.

The coatings reveal full, dense microstructures, such as those observed with the Al substrate. No visible connective porosity through the coating from the top to the interface coating/substrate can be observed. The SiC particles and a small quantity of pores are distributed homogeneously throughout. The Al 5056 particles are fully deformed as their original spherical shape can no longer be identified. The SiC particles retain their irregular morphology as before the deposition. There is no fracture of these particles as we had observed for bigger hard particles [30]. At the interface, there are neither cracks nor delamination.

Figure 3 presents the coating porosity and microhardness for the studied coatings. The addition of SiC particles significantly increases the coating density and microhardness. The porosity of the Al 5056 coating is more than 2.5%; in contrast, the porosity of the composite coatings is less than 1%. Meanwhile, the addition of the ceramic particles increases the microhardness from 110 $HV_{0.3}$ to 150 $HV_{0.3}$ (Comps1) and 175 $HV_{0.3}$ (Comps2). This is in good agreement with the results obtained with the coatings tested on the Al substrate [32].

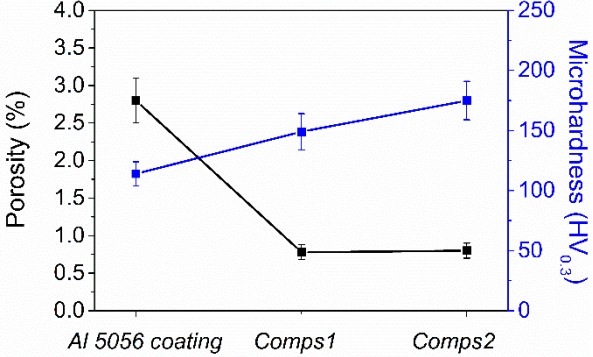

**Figure 3.** Porosity and microhardness results of the cold-sprayed Al 5056 coating and $SiC_p$/Al 5056 composite coatings on the magnesium substrate.

### 3.2. Corrosion Behaviour

The OCP values for the Al 5056 coating and the $SiC_p$/Al 5056 composite coatings, as well as the bare Mg substrate, are shown in Figure 4a. It is clear that all coatings exhibit much nobler potential values than that of the Mg substrate. Moreover, after reaching a steady state, the composite coatings exhibit a higher potential than that of the Al 5056 coating.

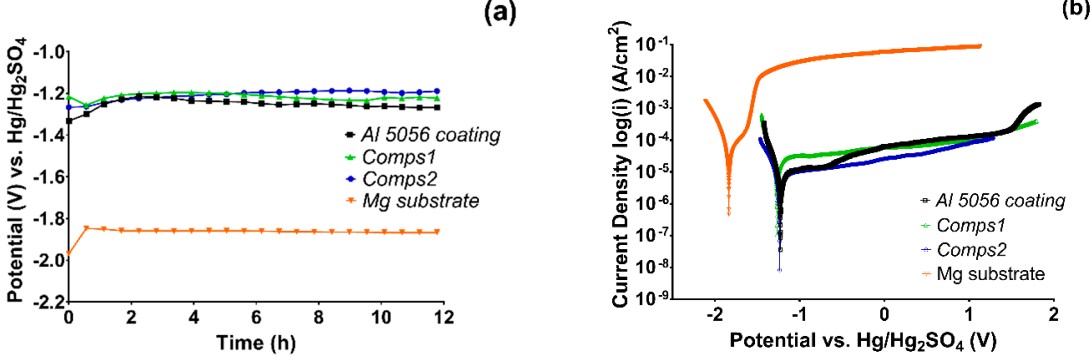

**Figure 4.** Electrochemical behaviour for the cold-sprayed Al 5056 coating, $SiC_p$/Al 5056 composite coatings and magnesium substrate in 0.1 M $Na_2SO_4$ solution: (**a**) open circuit potentials and (**b**) potentiodynamic polarization curves.

The results of the potentiodynamic polarization tests reveal that the Al 5056 coating and composite coatings exhibit similar polarization behaviour, which is very different from the curve of the Mg substrate (Figure 4b). Compared to the Mg substrate, the coatings show lower corrosion current density

($I_{corr}$) and higher corrosion potential ($E_{corr}$). This suggests that the cold-sprayed Al coatings can exhibit cathodic behaviour with respect to the Mg substrate. Furthermore, the composite coatings present higher corrosion potential ($E_{corr}$) and lower corrosion current ($I_{corr}$) than those of the Al 5056 coating without SiC.

The galvanic interaction between the coatings and the Mg substrate was studied by continuously monitoring the galvanic potential ($E_{gal}$) and galvanic current ($I_{gal}$) over 24 h (Figure 5). To compare, the $E_{gal}$ and $I_{gal}$ between the coatings and the Al substrate are shown as well, which were obtained by coupling the pure Al substrate with the coatings deposited on the Al substrate. The electrochemical behaviour of the Al coating and SiC$_p$/Al 5056 composite coatings on a pure Al substrate was studied in a previous work [32]. The corrosion potentials of the Al substrate, Al 5056 coating, Comps1 coating and Comps 2 coating are −0.72, −1.28, −1.25 and −1.25 V vs. Hg/Hg$_2$SO$_4$, respectively [32]. The coating/Mg couples exhibit $E_{gal}$ values close to the corrosion potential of the Mg substrate (Table 1). The system is under cathodic control. In contrast, the $E_{gal}$ values of the coating/Al couples approach the corrosion potential of the coatings. In the initial half hour of immersion, the galvanic current increases and then decreases until after 6 h. In the remaining test time, the galvanic current increases gradually. The negative galvanic current ($I_{gal}$) with respect to the coating indicates that the coating is the cathode in the coating/Mg galvanic couple. In addition, for the couples with the composite coating, the galvanic potential ($E_{gal}$) is higher and the galvanic current ($I_{gal}$) is lower than those of couples with the Al 5056 coating.

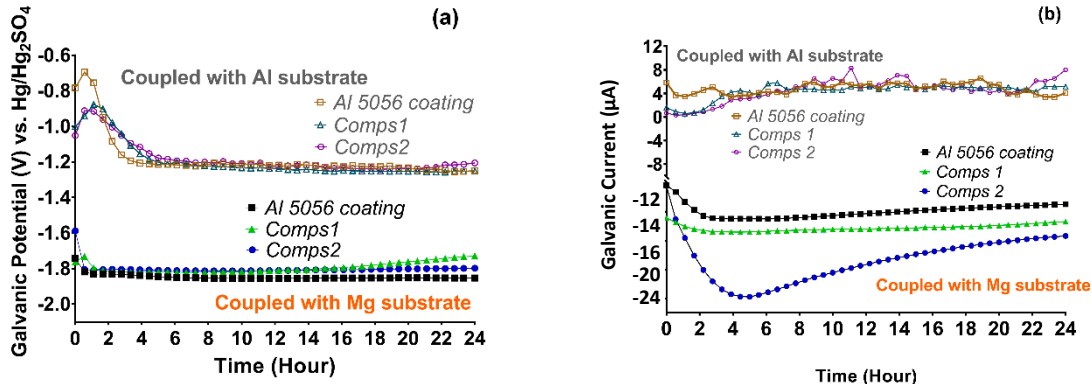

**Figure 5.** (**a**) Galvanic potentials ($E_{gal}$) and (**b**) galvanic currents ($I_{gal}$) of the coatings coupled to the Mg substrate and Al substrate in 0.1 M Na$_2$SO$_4$ solution.

**Table 1.** Electrochemical data of the cold-sprayed SiC$_p$/Al 5056 composite coatings and Mg substrate (all potentials are referenced to the Hg/Hg$_2$SO$_4$ electrode).

| Specimen | OCP (V) | $E_{corr}$ (V) | $E_{gal}$ (V) |
|---|---|---|---|
| Mg substrate | −1.87 | −1.83 | - |
| Al 5056 coating | −1.26 | −1.27 | −1.85 |
| Comps1 | −1.21 | −1.25 | −1.73 |
| Comps2 | −1.19 | −1.21 | −1.80 |

Figure 6 shows the effects of stirring on the galvanic corrosion of the Comps2/Mg couple. It can be seen that solution stirring has no obvious effect on the galvanic potential but decreases the negative galvanic current. Nevertheless, the tendency of the galvanic current to increase over time is not changed. The effect of stirring indicates that the galvanic corrosion of the coating/Mg couple is influenced by transport phenomena controlled by oxygen diffusion.

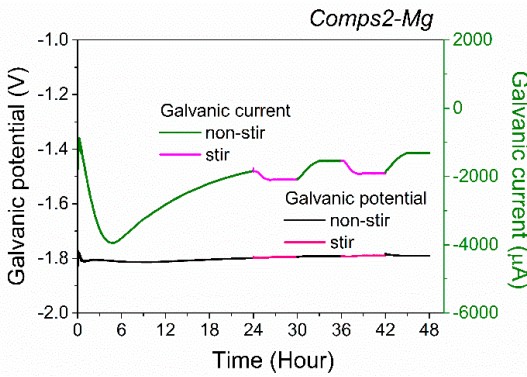

**Figure 6.** Effect of stirring on galvanic corrosion of Comps2/Mg couple.

Figure 7 shows the Nyquist and Bode plots of the cold-sprayed Al 5056 coating, the SiC$_p$/Al 5056 composite coatings and the magnesium substrate. The impedance spectra of the coatings mainly show semi-capacitive behaviour with several time constants, indicating that the electrochemical behaviour of the coatings exhibits several steps of dissolution and segregation of the corrosion products, which act as a protective layer. The diameter of the capacitive loop is noticeably larger than that of the Mg substrate. That demonstrates the beneficial effect of Al-base coatings on the corrosion protection. In addition, these capacitive loops are not perfect semicircles. This is attributed to the frequency dispersion effect associated to the kinetic of reaction, as a result of the surface roughness or the homogeneities in the coating, especially the distribution of SiC which exhibit a bad conductivity. Therefore, heterogeneities in the current line distributions and in the distribution of the resistivity are seen through the thickness of the corrosion product layers. These anomalies are characterized using a constant phase element (CPE) on the impedance diagram [33,34]. The impedance spectrum of the Mg substrate is very different. First, the diameter of the capacitive loop is much smaller. Second, in the low frequency range, there is an inductive loop, which goes from the first quadrant to the fourth quadrant. Inductive loops at low frequencies are a common feature in the corrosion of magnesium alloys [35,36]. This can be associated with the adsorption and desorption phenomena occurring on the surface of the sample, which leads to the formation of a corrosion product layer on the surface of the electrode. For example, in the case of magnesium corrosion, the inductive loop may be associated with the presence of Mg$^+$ or Mg$^{2+}$ ions resulting from the corrosion process and adsorbed on the magnesium surface [35].

Simultaneously, significant differences in the Bode plot can be found between the coatings and the magnesium substrate. The cold-sprayed Al-base coatings present similar impedances and phase angles. In contrast, the magnesium substrate shows a much lower impedance and a completely different phase angle. Considering the diagram obtained from composite coatings, it is important to highlight the fact that the semi-circle loops are similar and seem to increase (Figure 4a) and reach a phase angle ranged from 35° to 45° (Figure 4c). This behaviour illustrates a diffusional behaviour certainly through corrosion products, and a higher corrosion barrier effect of composite coatings. This assumption is confirmed with analysis of OCP during the long-term immersion.

Figure 8 shows the OCP evolution of the cold-sprayed Al 5056 coating and SiC$_p$/Al 5056 composite coatings on the magnesium substrate over an immersion period of 91 days. The OCP values of the coatings exhibit a transient phenomenon during the initial period, which indicates an evolution in the surface coating. On the curve of the Al 5056 coating, the OCP falls after 53 days, which corresponds to the coating degradation after an extended immersion. Similarly, the curve for Comps2 shows that the OCP falls after 77 days. The OCP of Comps1 shows three stages, similar to the aluminium substrate: i) from the beginning to the 1st day, the OCP decreases; ii) from the 3rd to 53rd day, the OCP fluctuates with the developing corrosion products; and iii) from the 65th to 91st day, the OCP remains stable as the corrosion products cover the coating surface. To confirm this point, EIS have been performed.

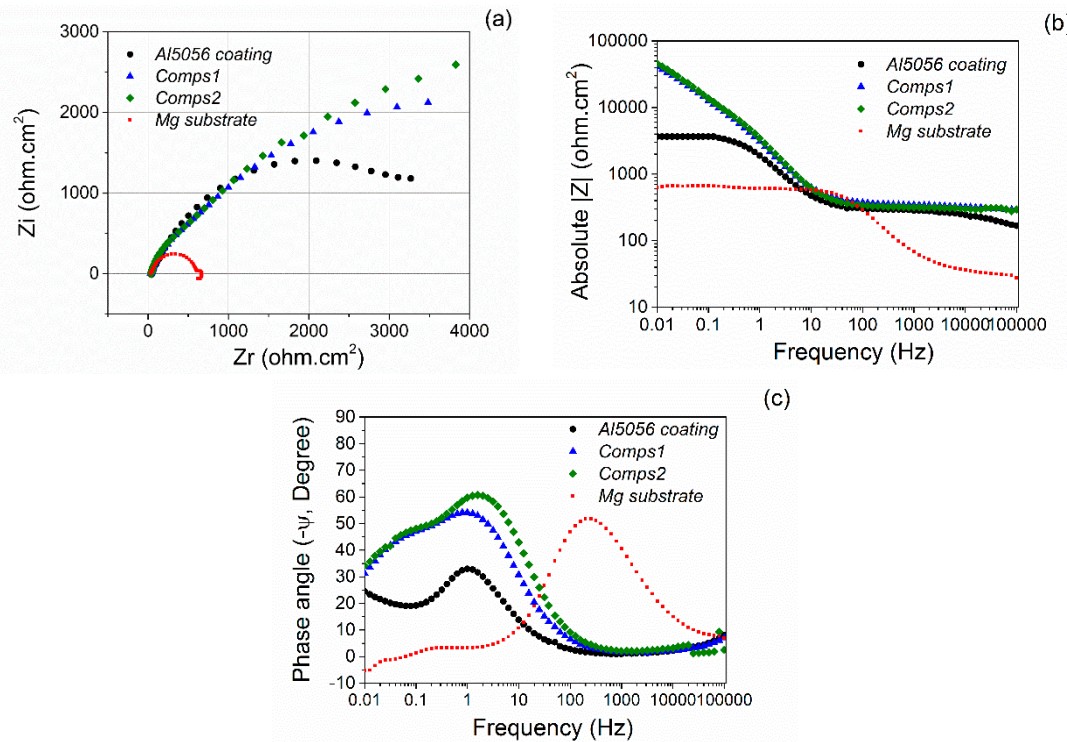

**Figure 7.** Electrochemical impedance spectroscopy (EIS) results obtained from the cold-sprayed Al 5056 coating, $SiC_p$/Al 5056 composite coatings and magnesium substrate: (**a**) Nyquist spectra and (**b**), (**c**) Bode plots.

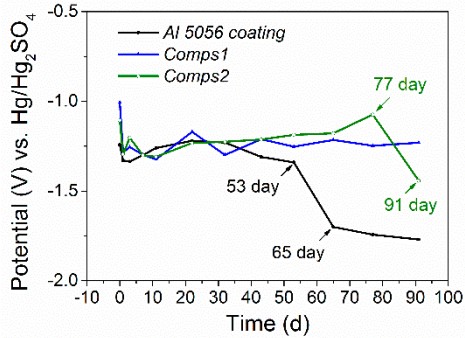

**Figure 8.** Open circuit potential (OCP) evolution of the cold-sprayed Al 5056 coating and $SiC_p$/Al 5056 composite coatings on the magnesium substrate for an immersion period of 91 days.

EIS was performed during the immersion period of 91 days. The Nyquist plots of the cold-sprayed Al 5056 coating on the magnesium substrate are shown in Figure 9. In accordance with the OCP evolution, from the beginning to the 53rd day, the impedance spectra mainly show capacitive loops, indicating the corrosion of the coatings. These capacitive loops are not perfect semicircles, which can be attributed to the frequency dispersion effect, as explained previously by the presence of the CPE [33]. Moreover, in this period, the impedance of the Al 5056 coating on the magnesium substrate shows a similar behaviour to the impedance measured with the Al 5056 coating on the aluminium substrate, which demonstrates the intrinsic behaviour of the coating. There is no effect of the substrate. In the initial immersion period (from the beginning to the 3rd day), the diameters of the Nyquist plot decrease. Furthermore, in the Bode plot, the impedance modulus decreases in the low-frequency region. After 7 days of immersion, the diameters of the semicircles start to increase and then fluctuate.

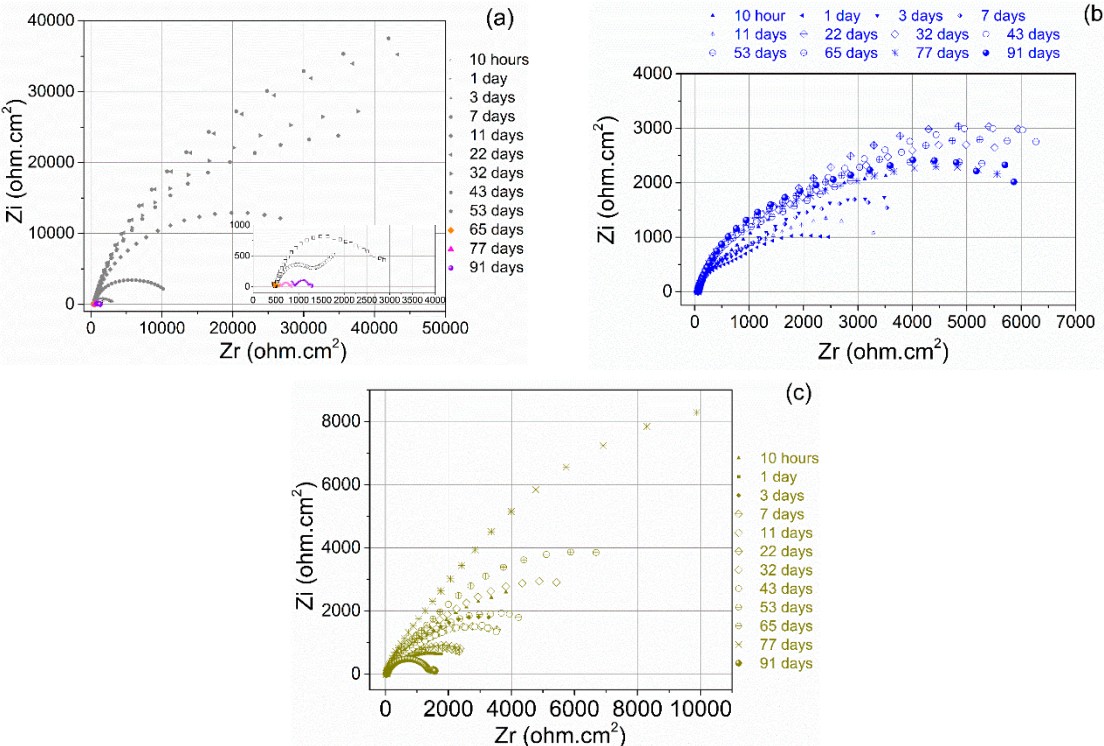

**Figure 9.** Impedance spectra (Nyquist plots) for the coatings after different immersion times in 0.1 M Na$_2$SO$_4$ solution: (**a**) Al 5056, (**b**) Comps1 and (**c**) Comps2 coatings.

After 53 days, the OCP values drop significantly with the impedance spectra. First, the diameter of the impedance diagram shows a relatively large decrease. In the Bode plots, a significant decrease in the impedance and a decrease and shift in the phase angle are observed. Second, an inductive loop appears in the low frequency range. These are the features of the magnesium substrate impedance.

Both the OCP and EIS results demonstrate that after 53 days of immersion, the solution reaches the interface between the Al 5056 coating and the magnesium substrate as shown in Figure 10 after 91 days of immersion.

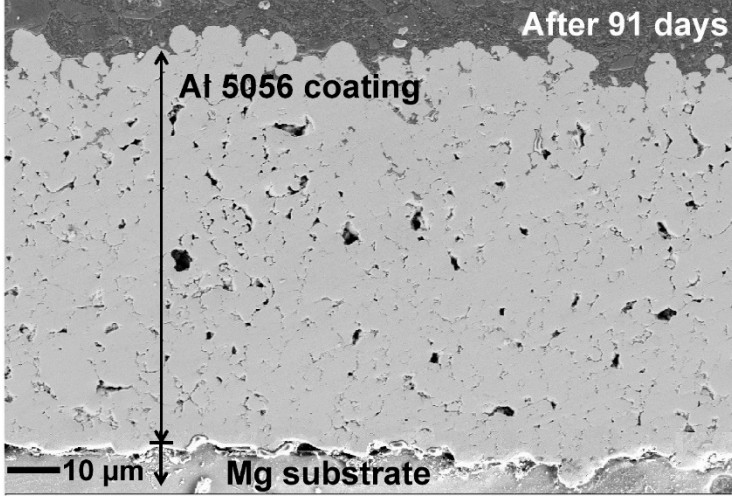

**Figure 10.** Cross-sectional SEM micrographs of the cold-sprayed Al 5056 coating on the magnesium substrate after immersion in 0.1 M Na$_2$SO$_4$ solution for 91 days.

On the Nyquist plot for Comps1 (Figure 9b), the diameter of the impedance varies slightly with time. This trend shows a stable behavioural pattern which is also characterized by a more stable OCP. The impedance values are high, which shows a resistance to corrosion, certainly due to the presence of corrosion products.

The impedance of Comps2 is in accordance with the OCP results. On the 77th day, the OCP increases, and the impedance shows its largest diameter. After 77 days, the OCP decreases drastically, the diameter of the impedance decreases and the shape of the phase angle related to the frequency curve changes.

The composite coating elaborated with smaller particle size exhibits a better behaviour in terms of corrosion resistance than the composite coating elaborated with the biggest particles.

After the immersion tests, cross sections of the Al 5056 coating were examined with SEM, as shown in Figure 10. After 91 days of immersion, the thickness of the coating decreased from 100 μm to 74 μm, and the porosity increased. Although the reduction in the coating thickness is not as large, corrosion has occurred on the magnesium substrate as the solution reached the interface. Therefore, the Al 5056 coating has no more protective effect since dissolution of the substrate occurs.

These results suggest that the lifetime of the coating cannot be determined from its dissolution rate. There is a strong possibility that the solution will reach the substrate before the dissolution of the coating; in this case, the substrate will be corroded preferentially which provides a risk of decohesion. First, due to the low corrosion potential of the substrate in the galvanic couple, its corrosion rate is much faster, promoted by the unfavourable surface ratio between coating and the substrate. Second, the solution follows the available pathways due to the open porosity of the coating, and crevice corrosion is promoted in narrow spaces. This is an important reason for the recommendation of the composite coating, even taking corrosion protection out of consideration.

### 3.3. Thermal Properties

The cross-sectional microstructure of Comps2 after 216 thermal cycles under condition I is shown in Figure 11. The sample is considered to be damaged as a crack wider than 100 μm in width is observed (Figure 11a). It is interesting that the crack is located in the Mg substrate but not at the coating-substrate interface in accordance with the distribution of the chemical elements (Figure 11b,c). The distance from the crack to the interface is approximately 160 μm. The morphology of the coating (area A) and substrate (area C) after applying the thermal shock conditions is obviously different. The layer (area B) above the crack seems to be similar to the substrate morphology at the microscopic scale. Current work is in progress to characterize the nanostructure which could correspond to a hardened zone.

Figure 12 shows an enlarged view of the rectangular area marked in Figure 11a. A layer that is approximately 70 μm in width and displays stretched features is observed. Area B in Figure 11 is identified as the substrate, since point No. 3 presents the same spectra as the substrate area. Consequently, the layer between areas A and B is identified as the location of the coating-substrate interface. Because this layer is rich in Mg and Al elements, it is reasonable to believe that this layer consists of Mg and AlMg intermetallic compounds. Interdiffusion of the Al and Mg occurs around the interface, and a hybrid Mg-Al compound is formed during the heat treatment [37].

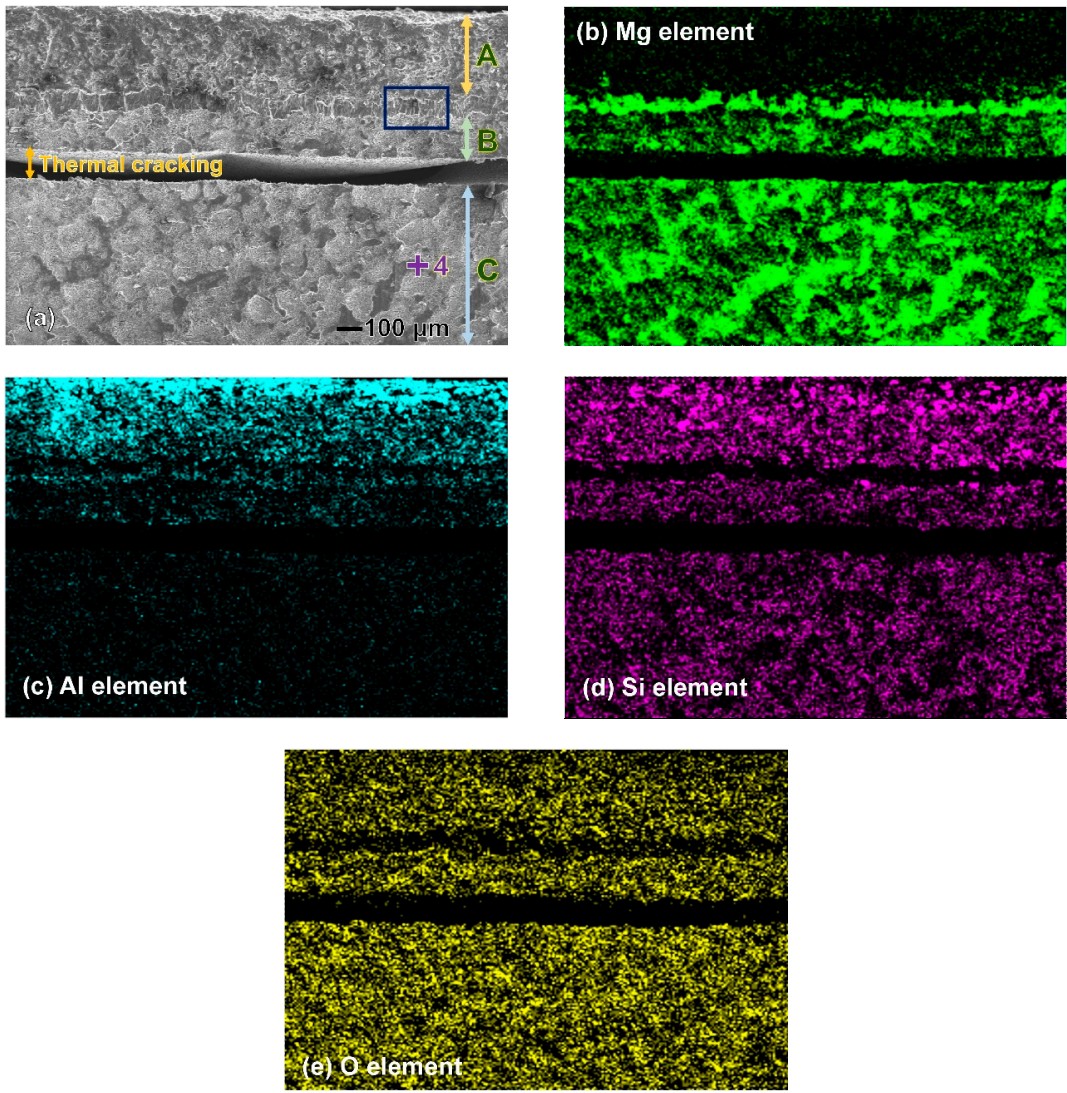

**Figure 11.** (**a**) Cross-sectional SEM micrographs, (**b**–**e**) and the corresponding elemental mappings (Mg, Al, Si and O Kα) of the damaged Comps2 coating after 216 thermal cycles in condition I.

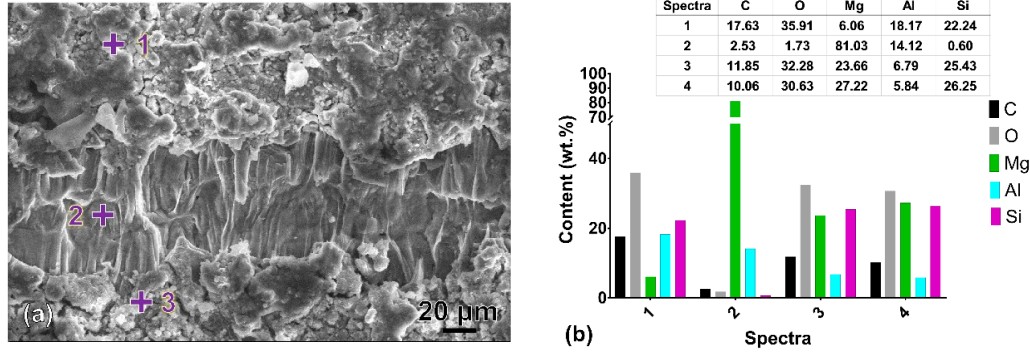

| Spectra | C | O | Mg | Al | Si |
|---|---|---|---|---|---|
| 1 | 17.63 | 35.91 | 6.06 | 18.17 | 22.24 |
| 2 | 2.53 | 1.73 | 81.03 | 14.12 | 0.60 |
| 3 | 11.85 | 32.28 | 23.66 | 6.79 | 25.43 |
| 4 | 10.06 | 30.63 | 27.22 | 5.84 | 26.25 |

**Figure 12.** (**a**) Enlarged view of the rectangular area in Figure 11a and the (**b**) elemental analyses for the spectra numbered 1 to 3.

The cross-sectional microstructure of Comps2 after 240 thermal cycles under condition II is shown in Figure 13. The sample exhibits several holes (Figure 13a), and it is quite reasonable to suppose that the holes would coalesce to form a crack after additional thermal cycles. As observed with condition I,

the damage location is in the Mg substrate but not at the coating/substrate interface. The thermal crack is located deep in the Mg substrate and is approximately 130 μm from the interface.

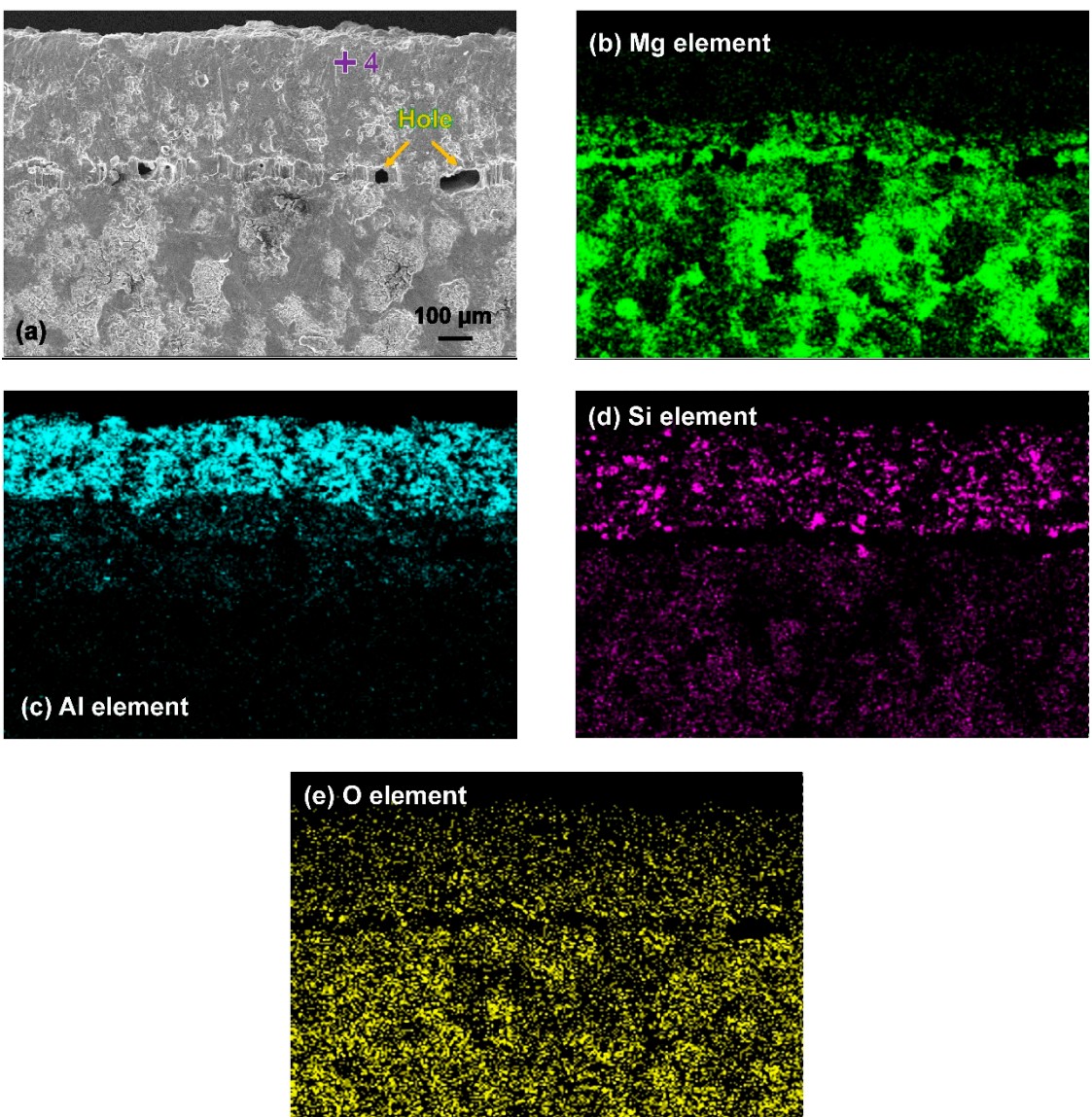

**Figure 13.** (**a**) Cross-sectional SEM micrographs and (**b**–**e**) the corresponding elemental mappings (Mg, Al, Si and O Kα) of the damaged Comps2 coating after 240 thermal cycles in condition II.

Figure 14 presents the enlarged view of a hole generated after the thermal test under condition II. The sample tested in condition II exhibits a remarkably low oxygen content compared to that in condition I, as the sample was heated in dimethyl silicone. The chemical composition of point No. 1 above the layer is very close to point No. 3. Both points reveal different compositions than that of the coating (point No. 4). A layer showing similar stretched features as seen in Figure 12a is observed, with a width of approximately 80 μm. This layer (point No. 2) is rich in Mg and Al elements. Interdiffusion also occurred in the aluminium-magnesium, even at 200 °C, which is relatively low according to the Arrhenius relationship [38]:

$$D = D_0 \exp\left(-\frac{Q}{RT}\right), \tag{1}$$

with the activation energy $Q = (31.15 \pm 0.28)\,\text{kcal}/\text{mol}$, (2)

$$\text{and the pre−exponential } D_0 = \left(1.24 \begin{array}{c} + \;\; 0.23 \\ - \;\; 0.21 \end{array}\right) \text{cm}^2/\text{s}. \tag{3}$$

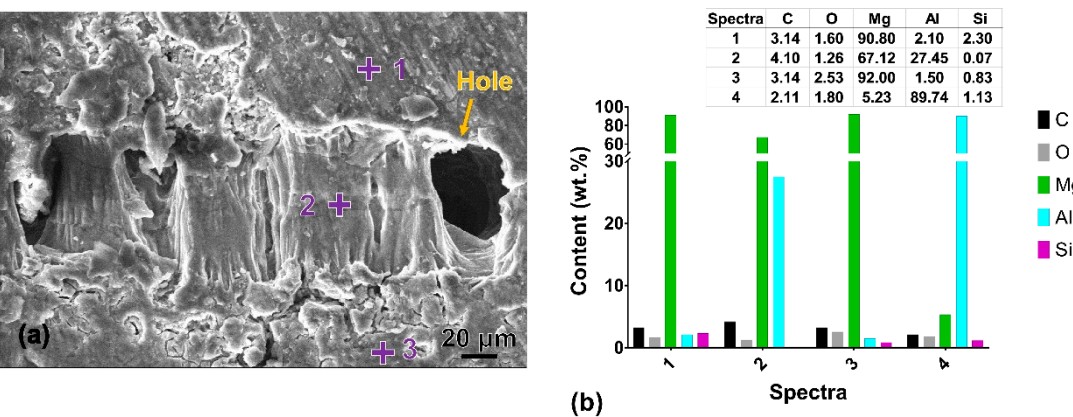

**Figure 14.** (**a**) Enlarged view of the thermal crack and (**b**) the elemental composition analyses of the spectra numbered 1 to 3.

The diffusion coefficient of Mg in Al based on the above equations is $6.39 \times 10^{-11} \sim 1.39 \times 10^{-10}$ cm$^2$/s, and $(3.09\sim7.99) \times 10^{-15}$ cm$^2$/s for the samples tested in conditions I and II, respectively. Therefore, aluminium diffuses more rapidly than magnesium [37]. The existence of oxygen in condition I greatly influences the interdiffusion of Mg and Al. Consequently, after more than 200 cycles, the impurity diffusion range of Mg and Al is comparable in the two thermal testing conditions.

From the above results, different testing conditions (heating temperature and ambience) only influence how many thermal cycles the samples endure. However, the location of damage does not change and is always on the side of the Mg substrate. The distances d$_f$ between the location of the damage and the coating–substrate interface are close to each other (160 μm and 130 μm) in both testing conditions.

In cold-sprayed coatings, a compressive residual stress can be expected as a result of plastic deformation. The residual stresses are very important factors that influence the microstructure of the coating–substrate system, as well as its performance. The process in which the material endures thermal cycles affects also the process in which the residual stress is relieved. Several researchers have found that residual stresses can contribute to enhanced fatigue life. [39,40]. Shayegan et al. [20] believe that the level to which crack growth is postponed depends directly on the size and depth of the compressive residual stress profile. The residual stress generated in an AZ31B substrate with particles of Al 1100 particles using different cold spraying parameters was studied by utilizing X-ray diffraction measurements and simulations [20]. The total residual stress depth is approximately 80 μm to 160 μm, which shows good agreement with the distance d$_f$ in this study.

Thus, a subsurface failure in the substrate means first of all that the bond between the substrate and the coating is strong. This is characterised by a chemical modification of the Al/Mg interface. This modification gives rise to a material of a different chemical nature whose composition depends on the thermal cycles undergone. However, the cracking occurring in the subsurface after some thermal cycling suggests that the subsurface of the substrate is worked down to a certain depth, leading to a microstructural modification of the substrate. It is therefore possible that the characteristics of the work-hardened substrate are different from the non-work-hardened substrate. In addition, the boundary between these two structures (work-hardened and non-work-hardened) is the stress accommodation location during thermal cycling. This accommodation is characterized by the initiation and propagation of the crack identified in Figure 12 and 13. Work is in progress to demonstrate these microstructure changes under impact and stress relaxation phenomena.

## 4. Conclusions

A dense aluminium coating and SiC$_p$/Al 5056 composite coatings are produced on magnesium substrates using cold spraying. The Mg substrate has no effect on the coating properties, such as microhardness and porosity. The addition of SiC particles results in a decreased porosity of the coating and better adhesion at the interface between the coating and the substrate.

In the case of galvanic coupling, the cold-sprayed aluminium coatings serve as a cathode for the magnesium substrate due to its high corrosion potential. In this case, the galvanic potential is close to the corrosion potential of the magnesium substrate, and the substrate is the anode. The addition of SiC particles increases the galvanic potential and decreases the negative galvanic current of the coating/substrate couple. Galvanic corrosion between the cold-sprayed SiC$_p$/Al 5056 composite coating and the magnesium substrate is influenced by oxygen diffusion since the negative galvanic current decreases due to solution stirring.

Monitoring the open circuit potential and electrochemical impedance spectroscopy in the long-time immersion tests illustrates that the addition of SiC particles improves the corrosion resistance of cold-sprayed coatings. Moreover, SiC particles with an average size of 15.6 μm reveal better improvement effects than SiC particles with an average size of 72.8 μm. If in the first configuration (15.6 μm) the corrosion resistance corresponds to coating properties, in the second case, the results suggest that the lifetime of the coating cannot be determined from its dissolution rate but from the interparticle dissolution and the dissolution of the substrate. Corrosion phenomena are governed by incubation period (progression of the solution within the coating) and galvanic corrosion of the substrate, providing a risk of coating debonding.

A cold-sprayed Al 5056 composite coating reinforced by 72.8 μm-SiC particles exhibits good properties in the thermal cycling tests. The samples endure more than 200 cycles before a fracture is observed. The testing conditions, including the heating temperature and ambience, influence the number of thermal cycles that the samples can endure. The observed location of damage is always deep in the Mg substrate and not at the coating–substrate interface. This result indicates that the crack propagation takes place on the subsurface, according stress relaxation between the work-hardened structure and non-work-hardened structure.

**Author Contributions:** Y.W., and B.N. conceived and designed the experiments; N.M., Y.W., G.Z. and H.L. performed the coatings; Y.W., B.N. and J.T. analyzed the data; B.N., and H.L. contributed materials; J.T. contributed analysis tools; and Y.W. and B.N. wrote the paper. All authors have read and agreed to the published version of the manuscript.

**Funding:** This research was funded by the National Natural Science Foundation of China (No. 51601158) and the Key R & D projects of the Science and Technology Department of Sichuan Province (No. 2018G20075).

**Conflicts of Interest:** The authors declare no conflicts of interest.

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
