# Peer review of "SiCp/Al5056 Composite Coatings Applied to A Magnesium Substrate by Cold Gas Dynamic Spray Method for Corrosion Protection"

_coatings, doi:10.3390/coatings10040325_

Round 1
Reviewer 1 Report
The critical point of this paper is that the advantages of adding SiC are not well highlighted. The only reliable information that can be obtained from the analysis of the experimental data is that the coatings containing SiC have significantly lower porosity values, probably due to a mechanical compaction effect on the Al matrix. Since the coatings are more compact, their behavior corrosion improves. This point should be better evidenced in the text of the paper: in the abstract, in the discussion, and in the conclusions. Unless the authors have an alternative explanation, which however should be explained just as clearly.
In the following some further remarks:
Process parameters used to deposit the coatings are missing and should be reported. The procedure used to determine the porosity contant shoul be described in the experimental part. What is the shock thermal behaviour of the coating that does not contain any SiC? In other word: has the SiC content any effect on it? May be, modifying the thermal expansion coefficient of the coating. Some statements in the conclusions are not justified by the experimental results, or at least need to be better justified:
4.1 “The addition of SiC particles results in a … better adhesion at the interface between the coating and the substrate.” This statement is not justified by any experimental result reported in the paper.
4.2 “The addition of SiC particles increases the 375 galvanic potential and decreases the negative galvanic current of the coating/substrate couple.” This is partially in contrast with the data reported in Fig. 5: no effect can be found on the galvanic potential, while there is an effect on the current, but only for Comps 2. Why? In the paper it is not discussed.
4.3 “…longtime immersion tests illustrates that the addition of SiC particles improves the corrosion resistance of cold-sprayed coatings.” The sentence is misleading; it seems that SiC particles have some “chemical” effect. Very likely, the better corrosion behaviour is a consequence of the lower porosity content, which depends by the hammering of the SiC particles on the Al matrix; i.e., it is an effect of an effect!
Reviewer 2 Report
Re.: Coatings, manuscript coatings-695449
Title: SiCp/Al5056 Composite Coatings Applied to a Magnesium Substrate by Cold Gas Dynamic Spray Method for Corrosion Protection
Authors: Yingying Wang, Bernard Normand, Hanlin Liao, Guofeng Zhao, Nicolas Mary and Junlei Tang
General Statement
The paper presents interesting results of a wide range experimental studies of properties and behaviour of SiCp/Al5056 composite coatings applied on magnesium substrate. The effort and time consuming investigations were focused on microstructure and anti-corrosion properties of two different composite layers. The properties were referred to properties of an Al5056 coating, i.e. coating produced with the same cold spraying technology with applying the basic component only. In general the manuscript is properly written and organized, the studies are thoroughly and clearly described, the outcomes provide original information of SiC powder addition to a spraying feedstock on the protective coating properties change. Except of a few typographical errors the manuscript could be recommended for publication almost at its present state. However, taking advantage of some non-substantive amendments necessary to suggest I am also submitting for discussion some questions. In my opinion answering them the authors could improve impact of their work.
Summarising, my recommendation is: accept with some editorial amendments and, if the authors agree, with some additional explanations or comments.
1. Selected Detailed Comments and Suggestions
In paper [30], to which the authors refer their work, the SiC powders of a average (vol.) diameter ranging from 2.3 to 72.8 mkm are presented. What was the basis of a choice of feedstock powders of average diameters 15.6 and 72.8 mkm? While comparing Figs 2.a and b one can notice inclusions of large SiC particles similar to those present in Comps 2 coating. In my opinion more diverse disperse structures could be obtained with applying SiC 4.7 mkm or 2.3 mkm feedstock powder addition. While was the expected effect of adding of SiC powder to the basic metallic feedstock powder? Was it just only expected coating hardness increase or the whole research was done in view of just cognitive gains. The SiC 72.8 mkm powder presented in [30] seems to be not “contaminated” by small diameter particles. What ids the reason of a distinct presence of small particles within Copms 2 structure (Fig. 2.c). Was it a fragmentation at collisions or impact responsible for this effect? At line 226 the authors attributed the discussed effects only to the coating surface roughness. In view of a non-homogeneous coating structure its effect on the results needs to be accounted for also – especially in view of a relatively bad electrical conductivity of SiC.
2. Typographical Problems
Denotation SiCp at lines 17, 23, 26 etc. needs to be corrected. In order to avoid a misinterpretation I suggest to exchange word “decreases” at line 214 with “intensifies”. Scaling of ordinate axis at Fig. 5.b needs the authors attention. Data presented in Figs 9.a, b and c, even at colour printing, are very hard to distinguish.
Reviewer 3 Report
First I want to point out that I am not a specialist of corrosion protection. In spite of that, I found that the paper is well written and presented. The different sections are clear and results are convincing. Thus, I recommend to accept this interesting and well written paper.
A
uthor Response
Please see the attachment

Round 2
Reviewer 1 Report
The authors answered to all my comments and I suggest to publish the paper.